# Feed Efficiency and Physiological Parameters of Holstein and Crossbred Holstein × Simmental Cows

**DOI:** 10.3390/ani13101668

**Published:** 2023-05-17

**Authors:** Deise Aline Knob, Armin Manfred Scholz, Laiz Perazzoli, Bruna Paula Bergamaschi Mendes, Roberto Kappes, Dileta Regina Moro Alessio, Ângela Fonseca Rech, André Thaler Neto

**Affiliations:** 1Centro de Ciências Agroveterinárias (CAV), Universidade do Estado de Santa Catarina (UDESC), Lages 88520-000, Brazil; 2Organic Farming with Focus on Sustainable Soil Use, Justus Liebig Universität—Giessen (JLU), 35394 Giessen, Germany; 3Lehr- und Versuchsgut Oberschleißheim, Tierärztlichen Fakultät, Ludwig Maximilians Universität München (LMU), 85764 Oberschleißheim, Germany; 4Núcleo de Educação a Distância, Centro Universitário Leonardo da Vinci, Rua Marechal Deodoro da Fonseca, Indaial 89084-405, Brazil; 5Estação Experimental de Lages, Empresa de Pesquisa Agropecuária e Extensão Rural de Santa Catarina (Epagri), Lages 88502-970, Brazil

**Keywords:** dry matter intake, heat stress, milk yield, rumination time

## Abstract

**Simple Summary:**

Optimizing feed efficiency (FE) is crucial for dairy farming as it enables better use of energy and reduces costs. Factors such as milk yield, diet composition, and genetic value can affect feed efficiency in dairy herds. Crossbred Holstein × Simmental cows have been shown to have better milk quality, fertility, and longevity, but no studies have compared their feed efficiency to purebred Holstein cows. With our study, we aimed to evaluate the FE and physiological parameters of Holstein and crossbred Holstein × Simmental cows to generate information that can help farmers make better decisions about the use of crossbreeding programs in their herds, especially regarding FE, which is a factor with higher economic impact. The results showed that crossbred cows had similar FE and milk production as purebred Holsteins, making them a viable alternative for high-production systems. Additionally, crossbred cows were found to be better at dissipating body heat in a heat-stress situation compared to purebred Holsteins.

**Abstract:**

This study aimed to compare the feed efficiency (FE) and physiological parameters of Holstein and crossbred Holstein × Simmental cows in a confinement system during winter and summer. The study was conducted in a dairy farm in southern Brazil by including a total of 48 multiparous cows. The cows were studied for 21 days in two periods, summer and winter, and their daily dry matter intake (DMI), milk yield (MY), rectal temperature (RT), respiratory rate (RR), body weight, and body condition score were recorded. An analysis of variance was conducted using the SAS statistical package. The results showed that crossbred Holstein × Simmental cows have a similar FE as Holstein cows in a high-production system (1.83 × 1.81 kg DMI/kg MY, respectively), and they can achieve the same production levels as purebred Holstein cows (43.8 vs. 44.5 milk/cow/day). Our findings indicated a difference for the period as both genetic groups achieved higher FE in winter than in summer (1.98 vs. 1.67 DMI/kg MY, respectively). In addition, we found evidence that crossbred cows are better at dissipating body heat during heat-stress situations, as they have higher RR in summer compared to purebred cows, while Holstein cows have higher RT in summer afternoons than crossbred cows. Therefore, using crossbred Holstein × Simmental cows is an alternative for high-production systems.

## 1. Introduction

Optimizing feed efficiency (FE) to achieve higher productivity in dairy herds with better use of energy is one of the most important issues in dairy farming. It enables better use of energy and reduces costs, as feed in dairy production is often the main cost in the production system [1]. It becomes so important that a breeding value for FE has been included in some breeding programs, e.g., in Australia [2] and in the United States of America [3], and the genes related to this trait have been studied for the Holstein and Jersey breed [4].

FE for dairy cows is usually estimated by calculating milk yield (MY) or fat + protein yield for each kilogram of dry matter intake (DMI) [5]. There is wide variation among farms, and numerous factors can influence the FE of dairy herds, such as MY and milk solids content, days in milk, diet composition and digestibility, body weight (BW) and body condition score (BCS), herd management, environmental conditions (especially heat stress), and cow breed and the genetic value of each cow [5]. For example, [6] compared the DMI and efficiency of two different breeds, Holstein and Simmental cows, and found that Simmental cows had lower DMI than Holstein cows, 17.59 vs. 19.49 kg/d, respectively. For MY, they observed a corresponding difference with values of 24.8 and 30.1 kg/d for Simmental and Holstein cows, respectively.

Crossbred Holstein × Simmental cows can improve milk quality, BCS, fertility, and longevity compared to Holstein dairy herds [7,8,9,10,11], without differences in calving difficulty and age at first calving between the genetic groups [7,12]. However, there are no studies comparing the DMI and FE of Holstein and crossbred Holstein × Simmental cows. In an evaluation of the first 150 days of lactation of Holstein and crossbred Holstein × Montbeliarde cows, there was no difference in MY and DMI between genetic groups [13]. Since BCS is an important factor affecting DMI and FE as well as MY, while crossbred Holstein × Simmental cows have a higher BCS than Holstein cows [7,8], it is necessary to mention that crossbred cows usually yield similar or only slightly lower milk yields than purebred Holstein cows [8,14,15,16,17,18].

Therefore, another important factor influencing FE for milk synthesis is the genetic composition of cows. Crossing Holstein with the dual-purpose Simmental breed may result in an altered allocation of feed energy between milk synthesis and deposition of body fat and/or body protein, which consequently may reduce FE for milk synthesis [19]. Based on these metabolic relationships, we hypothesize that the crossbred Holstein × Simmental cows will have higher DMI, similar MY, and lower FE compared to Holstein cows. Therefore, the objective of this study was to compare the FE and physiological parameters of Holstein and crossbred Holstein × Simmental cows in one confinement system during the winter and summer periods.

## 2. Methodology

All procedures used in this study were approved by the Ethics Committee of the Santa Catarina State University Ethical Committee, Protocol No. 6330030517.

The study was conducted in a commercial dairy farm located in the state of Santa Catarina, southern Brazil. The region belongs to a subtropical humid climatic zone, type Cfb, according to the Köppen classification [20]. The cows were kept in a compost-bedded pack barn. A total of 48 multiparous cows participated in the study, including 25 Holstein and 23 first-generation crossbred Holstein × Simmental cows. The study was conducted in 2 periods: (a) in summer (February) with 22 cows and (b) in winter (July) with 26 cows. Each study period lasted 25 days, including 4 days for cows to adapt to the experimental routine and the new group composition and 21 days for data collection. During the study period, the cows were fed the same diet as before the start of the study. To standardize the experimental groups, cows were always selected from the same farm management group (group 1 = high-yield group). During the summer period, the average number of days in milk (DIM) of the experimental group at the beginning of the study was 144 ± 75.8 days, while during the winter period, the average DIM at the beginning of the experiment was 96 ± 49.9 days. Only multiparous cows were used for the study. In both summer and winter, the average parity order for Holstein cows was 3.4 and 3.5, respectively. For the crossbred Holstein × Simmental cows, the average parity order was 3.5 and 4 in the summer and winter periods, respectively.

The feed offered to the cows was a total mixed ration (TMR) based on corn silage, ryegrass (fresh and as silage), and concentrates. The composition of the diet in winter and summer is shown in Table 1. The feed offered to the cows was calculated to meet 100% of the nutrient requirements [21]. Cows were mechanically milked 3 times per day (05:00 h, 13:30 h, and 20:30 h), and individual milk yield (MY) was recorded electronically (DeLaval^®^, Tumba, Sweden).

Individual milk samples were collected every 7 days in 40 mL bottles containing Bronopol [22]. Each sample consisted of an average mixture of the 3 daily milkings and was sent to the milk analysis laboratory of UDESC/Lages, SC, Brazil. The milk samples were analyzed for milk composition using a DairySpec (Bentley^®^, Chaska, MN, USA) instrument with the infrared method [23]. For each cow, an additional composite milk sample was collected on the same day in 40 mL bottles containing Bronopol^®^ as a preservative and then sent to a laboratory participating in the Brazilian Network of Quality Milk of the Brazilian Ministry of Agriculture, Livestock and Food Supply (MAPA) to perform somatic cell count (SCC) analysis using automated equipment (SomaCount FC and Dairy Spec FT Bentley^®^, Chaska, MN, USA) [24]. After milking, cows had access to the feed stall for approximately 2 h and 30 min. The feedlot had an individual, self-closing feed front where each cow stayed for the duration of each meal. The TMR offered was weighed and offered individually to measure individual feed intake. The basal feed mixture was offered ad libitum to each cow, allowing for 5–10% residual [25], and was prepared using a horizontal feed mixer. After each meal, the leftover feed was weighed. Samples were taken of the TMR offered and the residues from each cow, as well as the individual components of the feed, and then dried in a forced-air oven at 55 °C for 72 h. Samples were then crushed through a 1 mm sieve for subsequent chemical analyses [26]. Dry matter content was determined by drying the samples at 105 °C for 24 h. Ash was quantified by combustion in a muffle furnace at 550 °C for 4 h, and organic matter was quantified by mass difference. Total nitrogen was determined by the Kjeldahl method [26]. Neutral detergent fiber (NDF) concentrations were determined according to [27], except that samples were weighed into filter bags and treated with a neutral detergent in an Ankom A220 instrument (Ankom Technology, Macedon, NY, USA). Acid detergent fiber (ADF) and ADL concentrations were analyzed according to AOAC International (1998). Body condition score (BCS) was evaluated, and cows were weighed on the first and last day of each period. BCS was scored using a scale from 1 (extremely thin) to 5 (very fat) [28]. Cows were weighed in the morning after milking before their first meal. Daily rumination data were collected using the Heatime^®^ system (SCR/Allflex, Netanya, Israel), an automated system consisting of a collar with a tag that records the rumination time (in minutes) of each cow [29]

### 2.1. Physiological Measurements and Blood Collection

Every day, from the first to the fifteenth day of the experiment, we measured the temperature in the feed parlor with a digital thermometer (Kasvi^®^, São José dos Pinhais, Brazil) in the morning and in the afternoon. In the morning, around 6:00 am, and in the afternoon, around 3:00 pm, we measured the respiratory rate (RR) of the cows by observing coastal movements for one minute. After the RR measurements, we recorded the rectal temperature of each cow manually with a clinical thermometer (G Tech^®^, Duque de Caxias, Brazil) inserted carefully into the rectum with minimal disturbance to the animals.

On the last day of each experimental period, we collected blood samples from each cow by jugular vein puncture using a vacuum system with plastic tubes. Each tube contained a clot activator. After 3 h, the blood samples were centrifuged at 3000 rpm for 10 min for serum separation [30]. The serum samples were then frozen at −20 °C for later analyses. For the analysis of blood parameters, we sent the serum samples to the Laboratório de Patologia Clínica da UCEFF—Unidade Central de Educação FAEM Faculdade, Itapiranga, SC (Clinical Pathology Laboratory of UCEFF). Determination of total protein, albumin, and gamma-glutamyl transferase (GGT) was performed colorimetrically using specific commercial test kits (Gold Analisa Diagnóstica^®^, Belo Horizonte, Minas Gerais, Brazil). Colorimetric evaluations were performed using a semi-automatic biochemical analyzer (Bioplus 200S, Barueri, São Paulo, Brazil). For the determination of ketone bodies in blood (beta-hydroxybutyrate—BHB), we used the Optium Xceed portable BHB meter (Abott, São Paulo, Brazil).

### 2.2. Statistical Analysis

To obtain data normality, SCC was transformed into somatic cell score (SCS) by applying the following equation: SCS = log2 (SCC/100,000) + 3 [31]. To calculate milk yield corrected to 3.5% fat, we used the following equation: fat-corrected milk yield (FCM) = (0.432 × milk yield) + (0.1625 × milk yield × fat). Energy-corrected milk yield (ECM) was determined by the following equation: ECM = (0.327 × MY) + (12.95% × F × MY/100) + (7.65% P × MY/100), where MY = milk yield in l/day, F = fat percentage, and P = protein percentage [32].

Analysis of variance was performed using the procedure MIXED of the SAS statistical package [33] after testing the data for normality of residuals using the Kolmogorov–Smirnov test. The REML model included the fixed effects genetic group (Holstein, Holstein × Simmental), period (winter, summer), day (nested within the period), and the interaction between genetic group and period. The significance level was set at *p* < 0.05.

## 3. Results

The descriptive statistics (Table 2) of the population shows that there was no difference between the genetic groups for variables that could affect feed efficiency (FE), such as MY, milk content, and DIM, by the start of the research.

Crossbred Holstein × Simmental cows had similar FE (kg 3.5% fat-corrected milk/kg DMI) to Holstein cows (Table 3). The effect of the period was significant, with higher FE during winter. Both genetic groups showed similar MY, including ECM. However, there was a difference between periods, with higher MY and ECM in winter. As with MY, DMI showed no difference between genetic groups. In winter, cows consumed about 2 kg/day more feed (DMI) than in summer. For the variables fat and protein content, there were no differences between genetic groups and between periods. The same was observed for SCS, with both genetic groups having similar values. During the experimental period, cows had a daily rumination time of about 9 h, corresponding to about 37% of the day, with no differences between genetic groups.

Crossbred Holstein × Simmental cows were heavier and had a higher BCS than Holstein cows. This difference was observed in both periods. It should be emphasized that despite the high milk yield of both genetic groups, the cows did not lose weight or BCS from the beginning to the end of each experimental period (Table 3).

We did not observe any difference between the genetic groups for any of the blood parameters evaluated (Table 4). The observed values for total protein, albumin, globulin, albumin/globulin ratio, and GGT for both genetic groups were close. Although not significant (*p* = 0.083), the season seems to have a small effect on total protein, with higher values in winter compared with summer (8.37 ± 0.07 g/dL and 8.16 ± 0.09 g/dL, respectively).

For physiological parameters, we found only slightly higher rectal temperature (RT) in Holstein cows (*p* = 0.0624, Table 5), whereas RT differed significantly between periods with higher values in summer (*p* < 0.0001). The interaction effects were significant. Considering the afternoon evaluation as the hottest time of the day, we found that there was no difference between genetic groups in winter, but in summer, Holstein cows had 0.3 °C higher RT than crossbred Holstein × Simmental cows. The average values for respiratory rate (RR) showed that crossbred Holstein × Simmental cows had higher values than purebred Holstein cows. There was a significant difference between periods (*p* < 0.0001). In summer, cows doubled their RR. We also observed a tendency for interaction between genetic groups and periods. In winter, there was no difference between genetic groups, but in summer, crossbred cows had a higher RR (*p* = 0.057).

## 4. Discussion

This study aimed to investigate the FE and physiological parameters of Holstein and crossbred Holstein × Simmental cows in a highly intensive production system during the winter and summer periods. To our knowledge, this is the first study that investigates the feed efficiency of crossbred Holstein × Simmental cows in comparison to Holstein.

The results presented in Table 3 show the high productive performance and FE of crossbred Holstein × Simmental cows and Holstein cows in a confinement system in southern Brazil. These results are important to show that crossbred Holstein × Simmental cows have the potential to reach the same productive performance as Holstein cows without differences in FE, milk yield, ECM, and DMI. Therefore, our results can show that crossbred Holstein × Simmental cows are a suitable alternative for use in confinement systems where cows should achieve high productive performance to remain competitive in the system. In a study comparing the efficiency of crossbred cows in three different production levels, [34] showed that crossbred cows could be as competitive as Holstein cows in all production levels. For example, the Nordic Red × Holstein crosses produced the same or higher fat yield combined with better reproductive performance than the Holstein cows [34].

When comparing DMI, milk yield, and FE of purebred Holstein and Simmental cows throughout lactation, [35] reported that Simmental cows had lower DMI (20.2 vs. 21.8 kg/day), milk yield (27.4 vs. 35.5 kg/day), and FE (1.35 vs. 1.62 kg/day), respectively. Purebred Simmental cows produced about 77% of the milk yield of Holstein cows, while DMI reached 92%, resulting in a reduction in FE. When evaluating crossbred cows compared to purebred Holstein cows, we demonstrated that the differences seen in purebred cows were no longer present in high-performing Holstein × Fleckvieh crossbred cows kept in a confinement system, likely due to positive heterosis effects and complementarity between these two breeds. Crossbred Holstein × Simmental cows achieved similar FE to Holstein cows for milk synthesis regardless of the season. Additionally, [36], evaluating FE of Holstein cows, reported that cows whose rumen pH met the recommended values (pH ≥ 6.0) had a DMI of 26.8 kg/day, while the FE for milk synthesis was 1.94 (kg/kg), which are very similar values that we also found in our study during the winter season for both genetic groups. Thus, the feed offered to the cows seems to be sufficient to maintain stable rumen health and to meet the energy needs of the cows to reach their full production potential. In another study evaluating the FE of purebred Simmental cows at the beginning of lactation, [37] reported that the cows achieved a FE of 1.56 kg/kg while producing 36.1 kg milk/day and a DMI of 23.1 (kg/day) at a lower efficiency level than the Holstein and crossbred Holstein × Simmental cows in the present study.

Other authors reported that there were no differences in DMI and milk yield when comparing Holstein cows to crossbred cows. For example, [38] reported no difference in DMI (16.6 vs. 16.5 kg/day) and milk yield (21.8 vs. 21.1 kg/day) when comparing Holstein and a group of crossbred cows (Holstein × Jersey or crossbred Holstein × Norwegian Red cows). Furthermore, [39], who studied Austrian Holstein and crossbred Simmental × Holstein cows (with Holstein gene content between 50% and 75%), found no difference between genetic groups in DMI (20.86 and 20.82 kg/day, respectively) and ECM (29.20 and 29.30 kg/day, respectively). On the other hand, [40] reported that crossbred Holstein × Jersey cows produced more milk, based on a lower DMI than Holstein, and [41] found a superior feed, fat, and protein efficiency in Holstein × Jersey cows. In general, DMI and ECM reached lower values than in the present study, possibly because the cows were kept on pasture, whereas the performance data represent the average of the whole lactation.

Although the differences between genetic groups for FE, milk yield, and DMI were not significant, we observed a significant difference between periods. In winter, cows produced 12 L more milk per day and had about 2 kg/day higher DMI resulting in better FE (1.9 vs. 1.65 kg/kg). However, the differences could be mainly related to the cows’ lactation stage during each period. DIM averaged 96 days in winter and 144 days in summer. The average DIM in winter was close to the time when cows have their lactation peak, about 50–60 days after calving [17]. At this time, cows have the highest milk yield and the best FE of the entire lactation. Even in summer, with an average DIM of 144 days, when cows are in mid-lactation, they still have a high FE. Another important factor is that DMI as a percentage of body weight was about 3.0 and 3.3% in both genetic groups, with no difference between them. These values are in accordance with the recommendations for cows with high milk yield [21].

In addition to the lactation phase, rising temperatures in summer can also have a negative effect on production performance. Rising temperatures stimulate the satiety center, which causes cows to reduce their DMI as a tool to reduce the metabolic heat generated by rumen fermentation [42]. This homeostatic mechanism leads to physiological changes that affect milk yield and, consequently, FE for milk synthesis [43]. The authors of [44] showed that the performance of Holstein cows was lower in summer than in winter in terms of milk yield (23.32 vs. 24.59 kg/day), feed intake (37.67 vs. 38.92 kg/day), and FE (1.61 vs. 1.58 kg/kg), respectively. High-yielding cows are severely affected by higher temperatures [42] because they begin to lose the ability to regulate body temperature [45]. When comparing purebred Holstein and Simmental cows with high and low yields, both genetic groups of high-yielding cows show negative effects of high temperature–humidity index (THI) on milk yield. However, Simmental cows reduced their milk yield to a lesser degree than Holstein cows, indicating better heat stress tolerance in Simmental cows [45].

Another reason for the higher milk yield and FE of the cows in winter is the fact that the cows were offered different feeds during these two periods. In summer, the cows were given corn silage as a source of roughage, while in winter, they were given silage and fresh ryegrass as a source of roughage. Preservation of forages, such as silage, results in quality loss, which affects the content and nutrients of the silage [46] and may indirectly affect the milk yield of the cows. During the winter and summer periods, the cows were fed differently in accordance with the farm’s routine and the available feed ingredients during each season. The farm is located in a subtropical region in southern Brazil, where various forages are produced throughout the year, depending on the prevailing temperature and weather conditions. During the summer, the farm produces maize for silage, whereas during the winter, the same areas are used to produce ryegrass, which serves as a roughage source, either as silage or offered fresh to the cows. Fresh ryegrass has good nutritional quality [47], which enhances milk production in cows and, therefore, also improves feed efficiency.

Crossbred cows exhibited higher BCS throughout lactation, which may lead to better reproductive performance in these cows [7,8]. However, to maintain a higher BCS compared to Holstein cows, crossbred cows do not have a higher DMI (Table 3). The better BCS seems to be closely related to the complementarity effect of the breeds used in the crossbreeding programs. Simmental—as a dual-purpose breed—has a higher BCS than the Holstein breed [8,39]. Since they have the same FE, milk yield, and DMI as Holstein cows, it seems likely that crossbred Holstein × Simmental cows are more efficient in terms of energy use because, based on our results, they can maintain high production levels and still have a better (higher) BCS than Holstein cows. [48,49], who studied the energy efficiency of primiparous Holstein and crossbred Holstein × Jersey cows, concluded that the crossbred cows have similar or slightly lower energy intake. They require less energy for maintenance and use the same amount of energy for growth as the Holstein cows. However, crossbred cows produce the same amount of energy in milk, which means they are more efficient in terms of energy used to synthesize milk (solids).

BHB value is usually used as an indicator of fat/body reserve mobilization. Normally, the higher values occur at the beginning of lactation when cows have a negative energy balance and need to mobilize body reserves to meet the energy requirements for milk synthesis [50,51,52]. In our study, there were no differences between genetic groups with respect to BHB, which may be because the experiment was conducted after peak lactation. Despite the high milk yield, especially in winter, both genetic groups seem to obtain the required energy completely from the offered feed and are not forced to mobilize the body’s energy reserves. The authors of [53] found a difference between the genetic groups (0.45 vs. 0.51 mmol/L) when comparing the BHB of Holstein and a triple-crossed line Swedish Red × Jersey/Holstein. They evaluated the entire lactation and additionally compared low- and medium-production systems. Our values for BHB are higher than those reported by [53], possibly due to the higher production level of cows and the associated greater energy requirements for both genetic groups. In [54], wherein Fleckvieh and Holstein cows subdivided into high- and low-yielding cows were studied, authors reported a difference in BHB between the genetic groups with higher values for Holstein cows. Another interesting factor is that in the Simmental breed, the values for high- and low-yielding cows are very close (0.765 vs. 0.611 mmol/L), while in Holstein cows, the BHB values for high-yielding cows are much higher than for low-yielding cows (1.435 vs. 0.867 mmol/L).

Total rumination time and rumination time per kilogram DMI were similar in both genetic groups, possibly because DMI was also similar for Holstein and crossbred Holstein × Simmental cows. Total DMI is one of the factors related to rumination time. Other authors also found no difference in rumination time between Holstein and crossbred cows. The authors of [55] showed that there was no difference in rumination time between Holstein and Jersey cows. The same authors reported that rumination time has a positive correlation with milk yield (0.30). In general, cows with higher milk yield also have higher DMI, resulting in a negative correlation with rumination time (−0.14) [55]. Moreover, [56], comparing Holstein and crossbred Holstein × Jersey cows, found no difference between genetic groups for total daily rumination time at 426 and 383 min/day, respectively, and for rumination time per kg DMI (25.4 × 23.6 min). The authors of [57], however, showed higher rumination time for F1 and R1 crossbred Holstein × Jersey cows.

To examine indicators of immunity status related to mammary gland health, we evaluated some serum proteins in blood described by serum proteins, albumin, globulin, and albumin: globulin ratio [58]. In a previous study, crossbred Holstein × Simmental cows had lower SCS than Holstein cows (2.81 and 4.46, respectively) [17]. However, in the present study, we did not detect a difference in this trait between genetic groups. This may explain why we did not find a difference between genetic groups for serum proteins in the present study. The authors of [59] have shown that an increase in SCS also leads to an increase in total protein and albumin levels. The average values for total serum proteins obtained in our study are in line with the reference values for multiparous dairy cows for both genetic groups (7.4–9.2 g/dl) [60]. Additionally, [59] reported differences between specialized dairy breeds and dual-purpose breeds. However, we could not confirm this difference comparing Holstein and crossbred Holstein × Simmental cows, possibly due to high milk yield and similar SCS for both genetic groups.

Holstein cows showed a slightly higher rectal temperature (RT) than crossbred Holstein × Simmental cows (*p* = 0.0624), while crossbred cows achieved a higher respiratory frequency (RR) (*p* = 0.0310). This could be an indicator that crossbred Holstein × Simmental cows dissipate body heat more efficiently to maintain body homeothermy [61,62]. A similar observation was made by [43], where Jersey cows had a higher RR than Holstein cows, and the higher RR was associated with a compensatory mechanism to dissipate body heat [63]. The authors of [64] observed a negative correlation between RR and RT in Holstein cows, leading to the conclusion that the increase in RR was related to the decrease in RT. The higher RT in Holstein cows in summer, especially in the afternoon during the highest temperatures of the day [63], may be an indicator that this genetic group has more difficulty dissipating body heat than the crossbred Holstein × Simmental cows. Simmental, as a dual-purpose breed, has a different energy storage. Normally, beef breeds (and possibly dual-purpose breeds) convert more energy into muscle (protein), while dairy breeds store excess energy mainly as subcutaneous and visceral fat [65]. These fat compartments can act as thermal insulators and reduce endogenous heat exchange [66]. Moreover, [43,67] also reported that the deposition of subcutaneous fat negatively affects the thermoneutrality zone. Based on genetic complementarity, the dual-purpose Simmental breed might have an advantage in heat dissipation compared to Holstein cows.

Comparing the two seasons, we found that both genetic groups reached higher RR and RT values in summer than in winter (*p* < 0.0001). The authors of [63] reported similar results with higher values for RR (54.6 and 30 RR/min) and RT (38.6 and 37.8 °C) in summer than in winter. The thermoregulatory strategy of animals is to maintain a body temperature higher than the ambient temperature to allow body heat flux [68]. When the ambient temperature reaches values close to body temperature, the efficiency of body heat dissipation decreases [43,67,68]. This leads to a greater accumulation of body heat [69], and the only effective way to dissipate heat is to increase RR [43,62,68]. One limitation of our study was the lack of a datalogger installed at the commercial farm to measure not only temperature but also humidity. Therefore, we were unable to calculate the temperature–humidity index (THI). Furthermore, the data on the heat tolerance of crossbred cows are limited and require further investigation with a larger sample size to be confirmed. Despite these limitations, our study results indicate that crossbred Holstein × Simmental cows can be competitive with Holstein cows in a confinement system. This information is especially valuable for farmers seeking to benefit from the positive effects of heterosis, particularly for fertility traits, in Holstein herds. By using this crossbreeding program, they can achieve the same level of productivity and feed efficiency in their herds.

## 5. Conclusions

Crossbred Holstein × Simmental cows have similar feed efficiency to Holstein cows in a highly productive system. They can also achieve the same level of production as purebred Holstein cows. However, the period had a significant effect, with higher feed efficiency, milk yield, and dry matter intake during winter. These findings suggest that crossbreeding Holstein cows with Simmental bulls does not negatively affect their performance. We have found some evidence that F1 crossbred cows are better at dissipating body heat in a heat-stress situation, but further studies are needed.

## Figures and Tables

**Table 1 animals-13-01668-t001:** Ingredients and composition of the total mixed ration TMR ^1^ (% of dry matter) offered to the dairy cows during the research.

Ingredients	Summer Period	Winter Period
Corn silage	37.19	11.76
Ryegrass fresh	-	16.97
Ryegrass silage	-	9.40
Commercial concentrate	35.18	36.15
Brewery waste	8.99	9.15
Soybean hull	10.55	4.52
Ground corn	7.04	10.54
Mineral mix (commercial)	1.06	1.51
Chemical composition		
Dry Matter %	33.38	34.13
Organic material	95.24	91.81
Ash	4.76	8.19
Crude Protein	15.91	15.44
Ether Extract	3.50	5.04
NDF (neutral detergent fiber)	52.77	48.84
ADF (Acid detergent fiber)	22.73	17.36
NFC (non-fiber carbohydrates)	23.06	28.5

^1^ Values were obtained from chemical analysis of feed samples. NFC = 100 − (% CP + % NDF + % ether extract + % ash).

**Table 2 animals-13-01668-t002:** Descriptive statistics of the population of Holstein and crossbred Holstein × Simmental cows at the beginning of the experimental period.

Variable	Genetic Group			
	Holstein		Holstein × Simmental	
	Mean	SD	Mean	SD
Milk Yield kg/cow/day	44.3	8.5	45.06	8.72
Fat %	3.27	0.43	3.37	0.52
Protein %	3.06	0.26	3.07	0.17
Lactose %	4.92	0.17	4.83	0.28
Somatic cell score (SCS)	2.81	1.77	2.12	1.88
Days in milk	123.4	62.71	133.2	70.20
Parity	3.44	0.86	4.11	0.95

**Table 3 animals-13-01668-t003:** Least Squares Means (±SEM) and results of the variance analysis (*p*-values) for feed efficiency, dry matter intake (DMI), milk yield (MY), fat and protein content, rumination time (RT), beta-hydroxybutyrate (BHB), DMI % of body weight (BW) intake, BW and body condition score (BCS) for Holstein (Hol) and crossbred Holstein × Simmental (Hol × Sim) cows.

Variables		Genetic Group					Period				
		Hol		Hol × Sim			Winter		Summer		
	N	Mean	SEM	Mean	SEM	*p*	Mean	SEM	Mean	SEM	*p*
Feed Efficiency ^1^	1008	1.82	0.04	1.83	0.04	0.9065	1.98	0.04	1.67	0.04	<0.0001
DMI (kg/day)	1008	24.51	0.40	24.11	0.43	0.4492	25.49	0.44	23.14	0.48	0.0004
DMI % of BW	48	3.25	0.07	3.11	0.07	0.1529	3.32	0.07	3.04	0.07	0.0061
MY (L/day)	1008	44.5	0.92	43.8	0.98	0.5473	50.08	1.02	38.30	1.09	<0.0001
ECM ^2^	190	43.9	0.98	44.1	1.03	0.9118	49.7	1	38.4	1.1	<0.0001
FCM ^3^	190	43.06	1.03	43.23	1.08	0.9033	48.58	1.06	37.71	1.15	<0.0001
Fat %	190	3.34	0.06	3.33	0.07	0.9328	3.28	0.07	3.39	0.07	0.2786
Protein %	190	3.09	0.03	3.06	0.03	0.4913	3.07	0.03	3.08	0.03	0.9060
Fat + Protein (kg/day)	190	2.83	0.06	2.84	0.07	0.9385	3.19	0.06	2.47	0.07	<0.0001
Somatic cell score	189	2.8	0.30	2.16	0.31	0.1619	2.21	0.32	2.74	0.36	0.2517
BHB (mmol/L)	47	0.74	0.05	0.84	0.05	0.2404	0.70	0.05	0.87	0.06	0.0509
RT (min/day)	927	542	9.2	537	9.8	0.6231	560	9.8	519	10	0.0204
RT kg DMI (min/kg DMI)	927	22.5	0.6	22.7	0.7	0.8121	22.4	0.7	22.7	0.7	0.7432
BW (kg) first day ^4^	48	688	12	729	12	0.019	697	11	720	12	0.192
BW (kg) last day ^5^	48	699	11	739	11	0.017	722	11	716	12	0.754
BCS first day ^4^	48	2.79	0.08	3.62	0.09	<0.0001	3.31	0.08	3.10	0.09	0.101
BCS last day ^5^	48	2.90	0.09	3.66	0.10	<0.0001	3.35	0.09	3.21	0.10	0.334

^1^ Feed efficiency = L of milk/kg of dry matter intake; ^2^ energy-corrected milk yield; ^3^ fat-corrected milk yield 3.5%; ^4^ bodyweight (BW) and body condition score (BCS) on the first day of booth periods; ^5^ bodyweight (BW) and body condition score (BCS) on the last day of booth periods.

**Table 4 animals-13-01668-t004:** Means adjusted to the model ± standard error (SEM) of blood parameters for Holstein and crossbred Holstein × Simmental cows.

Variable		Genetic Group (GG)				*p*-Value		
		Hol		Hol × Sim		GG	Period	GG × Per
	N	Mean	SEM	Mean	SEM			
Total Protein (g/dL)	46	8.25	0.07	8.27	0.08	0.8475	0.0838	0.7980
Albumin (g/dL)	47	2.94	0.07	2.99	0.08	0.6627	0.8252	0.2821
Globulin (g/dL)	48	5.31	0.12	5.27	0.12	0.8402	0.5187	0.9080
Albumin:Globulin	47	0.56	0.02	0.57	0.02	0.6716	0.7757	0.3773
Gamma-glutamyl transferase (GGT) (U/L)	43	34.86	1.35	34.32	1.32	0.7793	0.6127	0.6430

**Table 5 animals-13-01668-t005:** Means adjusted to the model ± standard error (SEM) of the rectal temperature and respiratory frequency according to genetic group (GG) (Holstein or crossbred F1 Holstein × Simmental cows) and period (summer or winter).

Variable		Rectal Temperature		Respiratory Rate	
	Category	Means ± SEM	*p*	Means ± SEM	*p*
GG	Hol	38.43 ± 0.04	0.0624	45.66 ± 1.07	0.0310
	F1	38.29 ± 0.05		49.02 ± 1.12	
Period	Winter	38.08 ± 0.04	<0.0001	30.14 ± 1.06	<0.0001
	Summer	38.64 ± 0.05		64.54 ± 1.14	
GG × Period	Hol Winter	38.10 ± 0.06	0.1487	29.93 ± 1.51	0.0571
	F1 Winter	38.07 ± 0.06		30.33 ± 1.49	
	Hol Summer	38.75 ± 0.07		61.38 ± 1.52	
	F1 Summer	38.52 ± 0.07		67.69 ± 1.67	
GG × Period × Time	Hol Winter pm	38.18 ± 0.07	0.0154	32.90 ± 1.63	0.6143
	F1 Winter pm	38.17 ± 0.07		33.76 ± 1.60	
	Hol Summer pm	38.92 ± 0.07		64.02 ± 1.59	
	F1 Summer pm	38.64 ± 0.07		70.43 ± 1.74	

× = interaction between the variables.

## Data Availability

The data presented in this study will be made available on reasonable request from the corresponding author.

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
