# Peer review of "Feed Efficiency and Physiological Parameters of Holstein and Crossbred Holstein × Simmental Cows"

_animals, 2023, doi:10.3390/ani13101668_

Round 1

Reviewer 1 Report

Dear authors, your paper deal with Feed efficiency and physiological parameters of Holstein and crossbred Holstein x Simmental cows, interesting topic, very actual. However, I have some major concerns regarding the methods and included references. I suggest adding more references to the methods described to support the study. Moreover, digestibility analysis is missing in the paper. I suggest including all my suggestions before further consideration of the manuscript. In the attached file my specific comments.

Reviewer 2 Report

The manuscript is important in terms of providing valuable information about “Feed efficiency and physiological parameters of Holstein and crossbred Holstein x Simmental cows” The manuscript contains some syntax errors and misspellings. Revise the text to improve readability. I make some recommendations for improving the proposed paper. I also made some corrections to the text.

1. The introductory section could be improved.

2. Abbreviations used in the article should be explained where they were first used.

3. In the material, milk samples were taken, and somatic cell counts were measured, but they are missing in the Results. If it will not be used, this information should be removed from the material.

4. It is understood from the article that cows were offered different feed ingredients in summer and winter. Could this be the reason why some of your results are different? This situation should be explained better in the Discussion.

5. The conclusion can be improved.

Round 2

Reviewer 1 Report

The authors have responded to all the comments raised in the previous review. From my point of view, the manuscript can be published.